# Light Electric Vehicles for Muscle–Battery Electric Mobility in Circular Economy: A Comprehensive Study

**Sven Wüstenhagen [1,\*], Paul Beckert [1], Olaf Lange [2] and Andreas Franze [3]**

1   Fraunhofer IMWS, Walter-Hülse-Straße 1, 06120 Halle (Saale), Germany; Paul.Beckert@imws.fraunhofer.de
2   Olaf Lange Dreiradbau, Saarbrücker Str. 22-24, 10405 Berlin, Germany; Olaf.Lange@berlin.de
3   FVK GmbH, Am Waggonbau 3, 06844 Dessau-Rosslau, Germany; a.franze@fvk-dessau.de
\*   Correspondence: sven.wuestenhagen@imws.fraunhofer.de

**Abstract:** Light electric vehicles (LEVs) facilitate a significant reduction in global warming potential (GWP) and other environmental impacts related to specific transport performance due to their lightweight construction. Low-voltage systems in the drive engine, an open vehicle design and online vehicle data processing allow LEVs to be repaired by independent workshops, thus enabling long vehicle use as well as conversion or retrofitting for periods of use beyond 20 years. LEVs are not yet very common in everyday life in Western Europe. In order to support the acceptance of muscle power-supported LEVs in the EU L7e registration class by users, the vehicle design and construction specifically address requirements in the areas of last-mile parcel delivery and other transport services, including passenger transport. Life cycle assessment was used to investigate two construction methods for LEVs, mixed construction and fibre composite construction, in terms of the production, service life phase and end of life. A vehicle configuration was developed which, in addition to resource-saving production and long-life operation, enables easy access for users and maintenance of the LEV for various purposes. The resource efficiency of light electric vehicles was proven with regard to the ecological aspects. The vehicle design shown here shows high potential for LEVs in the circular economy.

**Keywords:** light electric vehicle; global warming potential; life cycle assessment

## 1. Introduction

The use of energy today, in the form of burned fuels in road-bound passenger and freight transport, is nearly constant in relation to 1995, with a slight increase of 7% in 2019 [1]. Life cycle assessment (LCA) studies on electric vehicles (EVs) show a correlation between vehicle equipment mass and environmental impact [2,3]. In addition to the use of electric drives, the lightweight construction offers approaches for reduced greenhouse gases with a moderate increase in manufacturing costs compared to conventionally configured road vehicles [4]. Further increases and the potential reduction in environmental impact have been identified for LEV freight transport in the "last mile" and trades [5]. Road-bound traffic density for the transport of goods and people in the "last mile" with conventional vehicles and conventional EVs has an increasingly negative impact on the quality of life in urban areas, due to particulate matter emissions from brakes and tires, noise and land use. As known from classical physics, the rolling resistance of a terrestrial vehicle is directly related to the mass, aerodynamics and rolling friction in the tires. Specific to EVs is a strong relation between vehicle mass and energy consumption, because of the good efficiency of electric machines [6].

The main subject of this paper is a comparison of two methods of construction for LEVs realized at prototypic scale, Cargo Cruiser 1 (CC1) with steel/GFRP construction and Cargo Cruiser 2 with GFRP construction. Table 1 shows the orientation by weight and energy consumption of LEVs by two possible construction methods, steel/GFRP and

GFRP construction, for one market-accessible EV and one market-accessible light-duty vehicle (LDV).

**Table 1.** Overview of construction methods considered for two LEVs, Cargo Cruiser 2 (CC2) with steel/GFRP construction and Cargo Cruiser 3 (CC3) with GFRP construction, in comparison to EV and LDV.

| | LEV with Steel/GFRP Construction (CC2) | LEV with GFRP Construction (CC3) | EV (Accessible on Market) | LDV (Accessible on Market) |
|---|---|---|---|---|
| Weight (kg) | 517.2 | 534 | 1200 | 1995 |
| Approximate energy consumption without payload (kWh/100 km) | 10 (electric) | 10 (electric) | 20 (electric) | 80 (diesel) |
| Silhouette in same scale |  |  |  |  |

To solve the issue of transporting goods and people in a compatible manner in the future, vehicles with low environmental impact and new models of transport have to be investigated in combination and this study approach proposes examples of vehicles in "last mile" applications. Increased services such as car sharing [7] shows the acceptance of new models of transport if they correspond to daily life practice. Multimodal transport systems for freight in urban areas are at an early stage of development [8] and for passenger transport, the needed integration of different modes is an open challenge, as shown by the low sustainability in the user availability of such offers [9]. Residents of urban centres solve this challenge in individual information work, provided they are prepared to be flexible and have the necessary knowledge to deal with information and means of transport [10]. Due to historically grown infrastructure and spatial traffic conditions, intermodal transport systems, for both goods and people, usually require solutions for the so-called last mile in order to successfully integrate individual and public passenger transport [11] and to make the transport of end user-related goods more efficient [12].

The study investigated the circularity of a last mile-specific LEV for the transport of people or goods. The LEV was developed with two methods of construction: steel/glass fibre reinforced plastic (named CC2) and nearly complete glass fibre reinforced plastic (GFRP, named CC3). The circularity of both constructions was investigated, with a focus on three aspects: production, maintenance and the concept of safety gained by lightweight construction. Because easy access to the vehicle is important, the vehicle was conceived as a muscle-electro-hybrid-powered vehicle, which comes close to the user experience of a heavy cargo bike. To reflect its usability as a vehicle for last mile transport, an intermodal model was used to describe the LEV's specifications. Since the planning effort required for intermodal transport in terms of logistics generates little additional expenditure compared to intermodal passenger transport, a life cycle assessment (LCA) comparison of the two methods of vehicle construction was focused on the last mile parcel delivery.

The results show that the production of light vehicles enables a significant reduction in greenhouse gases, as well as emissions of particulate matter and the demand for traffic space in their use phase. Involving small and medium enterprises (SMEs) in vehicle production seems possible and useful, e.g., when repairs are not limited to the use of original parts. The use of Long-Range Wide-Area Network technology was identified for improved safety and longevity of (light) vehicle technology and increased vehicle utility value.

## 2. Materials and Methods

The vehicle was developed using computer-aided methods. Two construction methods, mixed steel/GFRP and GFRP and the elaboration of the ergonomics of the driver (Figure 1a–c) were developed using CATIA V5 6R2020 software. For weight optimisation and fibre composite design, finite element models were used with ANSYS AIM 16.2 software and the Workbench PlugIn LS-PrePost® V4.7.7 [13]. Material parameters required for the component design were determined using the relevant material testing standards or taken from material data sheets, when available. The vehicle development was based on the guidelines of the EU L7e vehicle class and for this a pull-out test of the safety belts was undertaken (Figure 1d–f).

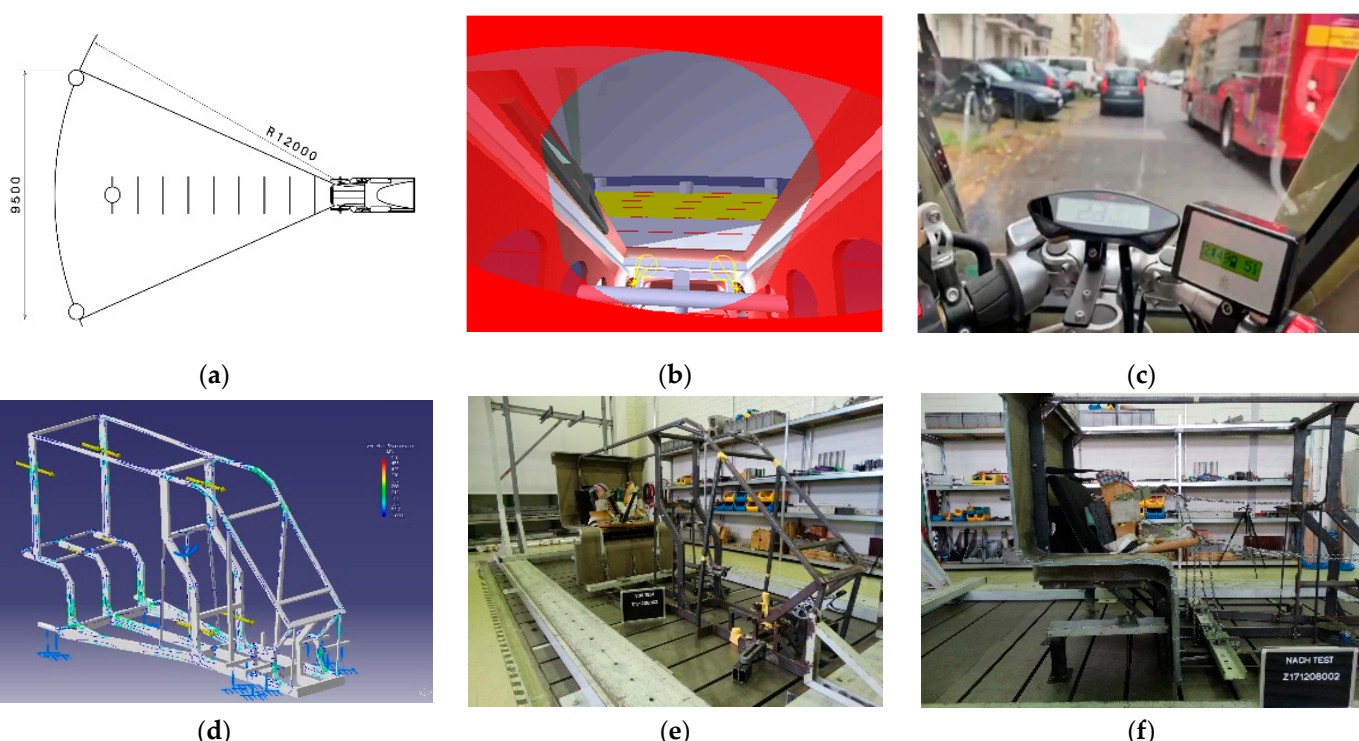

**Figure 1.** (**a**) Required field of view in front of a LEV. (**b**) Ergonomic model to pre-evaluate field of view for a tall person, represented by a mannequin for 95th percentile of Europeans. (**c**) Validating real-life view for average person from 50th percentile of Europeans via head-mounted dash camera. (**d**) Finite element model of steel frame (steel/GFRP construction) of Cargo Cruiser 2 showing stress strain (MPa) during load case of testing pull-out forces of rear safety belts. (**e**) Test bed for pull-out at front and rear seat safety belts for CC2. (**f**) After pull-out test of rear safety belts, no deformation in steel frame was observed, as expected, in FEM analysis.

To monitor the mechanical properties of the lightweight construction, a new type of sensors for detection of mechanical stresses, based on shape memory alloys (SMAs), was tested. The readout of sensors via an ESP32 microcontroller using a Long-Range Wide-Area Network (LoRaWAN) was evaluated during test driving.

In order to describe the repair capability as an orientation, classification of the light vehicle was carried out using the French repair index [14].

The life cycle assessment was carried out using Umberto LCA+ software from the IFU Institute Hamburg, Germany. The two construction methods to be investigated were designed for small-scale production by a company that manufactures fibre composite structures for aircraft and rail vehicles. For the life cycle inventory (LCI), generic datasets (secondary data) from the Ecoinvent 3.7.1 database were used for the traction battery and road use; the vehicle production data are balanced by material and energy flows (primary data) determined at the manufacturing plant. The entire life cycle (cradle to grave) of the

vehicles considered with the system boundary "German road network" was balanced in accordance with DIN EN ISO 14040 and the directives of the European Union [3,15].

The software and databases used are proprietary and available via licences.

## 3. Results

The results presented here are related, whereas the results of vehicle construction form the basis for all other results. The vehicle construction results provide the material flows in the production and disposal of vehicles and enable test drives to be carried out to determine energy consumption on the last mile and, thus, provides input for the investigation of reparability and component safety and LCA. The individual results were not used for iterative improvements and are presented here consecutively.

### 3.1. Vehicle Construction

The vehicle development met the specifications of a speed of 50 km/h, a payload of 250 kg in 3 m³ volume and the ergonomics for a muscle power-electric drive. The electric engine comprises 2 × 7 kW asynchronous motors near the rear wheels and a maximum of 100 W of muscle power from the driver. The vehicle is four-wheeled because of expected acceptance [16]. The dimensions of the Cargo Cruiser are as follows: overall length 3300 mm, width 1300 mm and height 1800 mm. To meet the ergonomic needs of all users of the muscle-electric vehicle, the driver's seat is an adjustable recumbent bike seat and the field of view is given. Two construction methods, steel/GFRP construction (steel frame and glass fibre reinforced plastic in the vehicle glider) and GFRP (reduced use of steel), were created for possible production in a fibre composite processing SME (Figure 2).

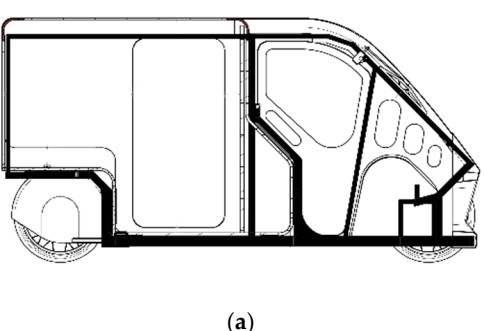 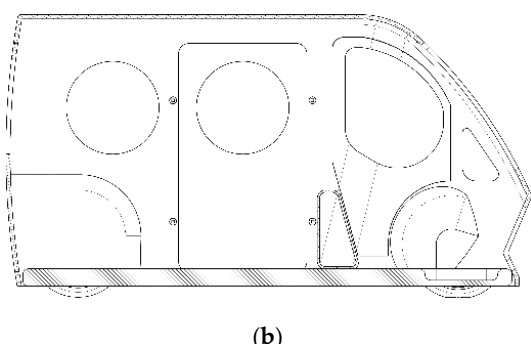

(**a**)        (**b**)

**Figure 2.** (**a**) Light electric vehicle Cargo Cruiser 2 (CC2) with steel/GFRP construction, with large mass fraction of structural steel (sectional view, black elements indicate4 steel frame). (**b**) Cargo Cruiser 3 (CC3) with GFRP construction, with minimised steel fraction. Hatched elements indicate sandwich construction with GFRP decking laces and PET foam core.

To measure the energy consumption of the LEV, test drives were performed under real-world conditions in Berlin, including 250 kg payload, in summer, with more than 90 start-stop situations per 100 km. The mass fractions resulting from the two construction methods were processed for life cycle assessment (Table 2).

### 3.2. Maintainance and Optimized Lightweight Construction

In order to determine the operational reliability of the GFRP construction, strain gauges were applied to the GFRP components when operating the Cargo Cruiser vehicle to enable structural health monitoring (SHM) for reliability tests and possible reduction in safety factors in lightweight construction. Pseudoelastic shape memory alloys (SMAs) are successfully used for this purpose in order to achieve a long service life of mechanical sensors [17,18]. The readout of SMA sensors during vehicle operation was tested by an ESP32 microcontroller.

For remote transmission of the measurement data, an energy-efficient Long-Range Wireless-Access Network (Figure 3) and the time-based InfluxDB database were successfully used on a test basis.

**Table 2.** Overview of construction methods considered for (**a**) CC2 light vehicle with steel/GFRP construction and (**b**) CC3 with GFRP construction. Higher vehicle weight with GFRP construction results from conservatively selected safety factors, as this construction has not yet been sufficiently evaluated in terrestrial vehicles.

| (a) | | |
|---|---|---|
| **Steel/GFRP (CC2)** | **m (kg)** | **Comment** |
| Components | 73 | |
| Steel | 277 | |
| GFRP decking layers | 19.2 | 50% wt Fibre |
| PET foam | 20 | Height: 15 mm |
| GFRP free-form parts | 22 | 50% wt Fibre |
| Battery | 106 | |
| Sum | 517.2 | |

| (b) | | |
|---|---|---|
| **GFRP (CC3)** | **m (kg)** | **Comment** |
| Components | 73 | |
| Steel | 7 | |
| PET foam | 48 | Height: <14 mm |
| GFRP free-form parts | 300 | 50% wt Fibre |
| Battery | 106 | |
| Sum | 534 | |

### 3.3. Independence from Original Parts and Repairabilty

According to the available parameters for representation of the French repair index, an introductory repair index for CC3 was determined based on the category "battery-powered lawn mower", as this product category highly corresponds with the CC3 light electric vehicle.

The design allows the use of third-party, non-manufacturer-specific components in the light vehicle. A very good score of the repair index was achieved under the assumptions made (Table 3).

### 3.4. Life Cycle Assessment

Both construction methods were successfully designed for small series production in a composite fibre processing plant and use identical drive and control technology. The battery system was defined as a unit by itself [19], in alignment with LCA practices [20]. Therefore, transferable starting assumptions (Figure 4) could be made to define the goal and scope for both designs and LCA comparability was achieved.

The functional unit is 1 tonne-kilometre (1 tkm) oriented on the logistics sector and represents a conservative assumption for LCA, but it allows a comparative study with generic datasets of two other vehicle classes: electric vehicle (EV), here in form of a battery-powered electric passenger car and light-duty vehicle (LDV). in the form of a van, both with an equal payload of 250 kg.

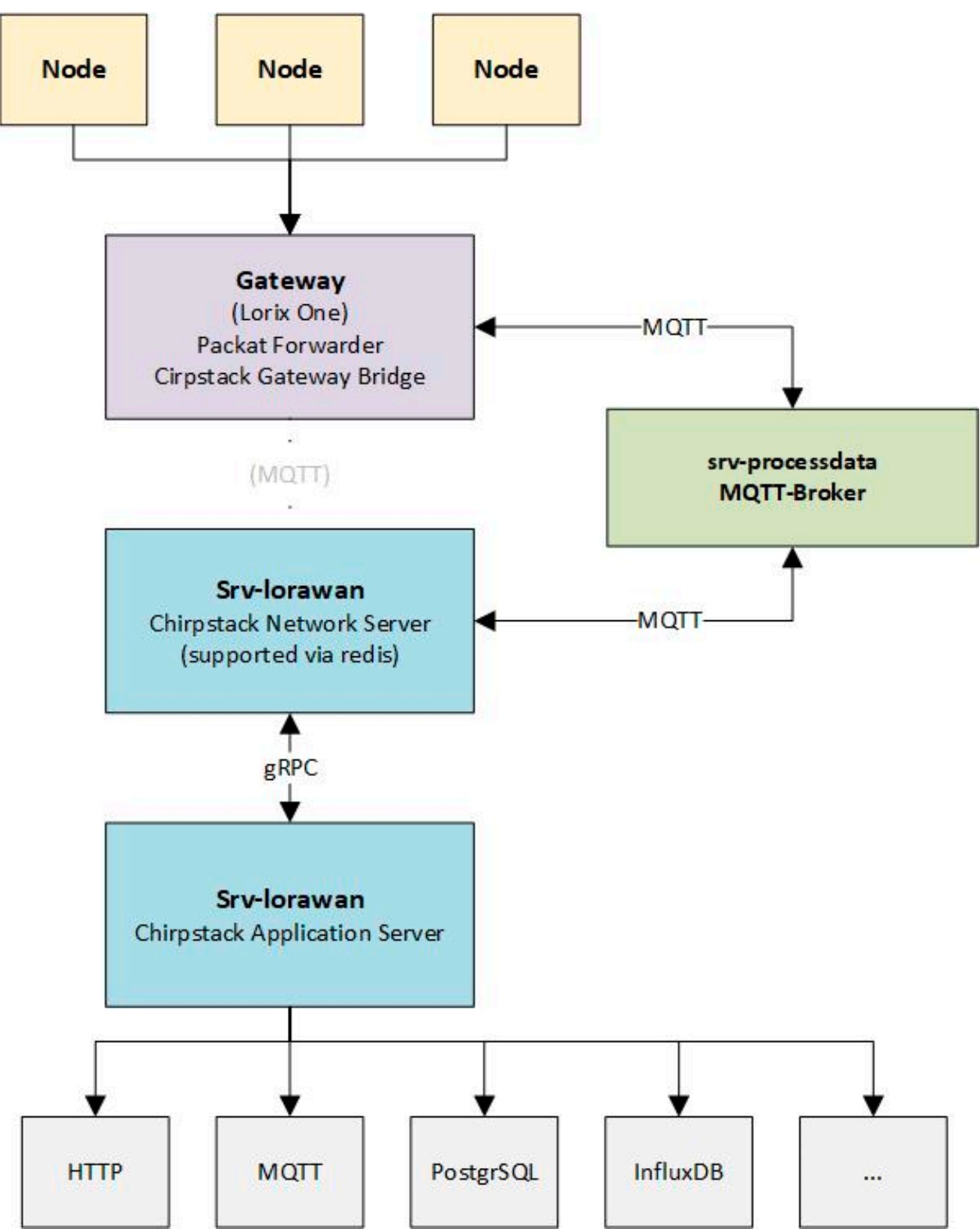

**Figure 3.** Schematic representation of Long-Range Wireless-Access Network (LoRaWAN) setup for energy-efficient acquisition of data on operational safety of fibre composite components in light vehicles.

**Table 3.** Parameters for repair index in product category "battery-powered lawn mower" and point entry from assumptions for initial determination of repair index for CC3.

| REPAIRABILITY INDEX CALCULATION AND PRESENTATION OF THE PARAMETERS WHICH ALLOWED TO ESTABLISH IT | | Cordless electric lawn mower (battery) | | | |
|---|---|---|---|---|---|
| Date of calculation | | 24 September 2021 | | | |
| Producer's or importer's name or trademark | | Please fill out | | | |
| Producer or importer adress | | Please fill out | | | |
| Producer's or importer's model identifier | | Cargo Cruiser III | | | |
| This "FINAL_SCORE" tab in English is purely indicative. In order to meet regulatory obligations, only the "NOTE_FINALE" tab in French | | | | | |
| Criteria | Sub-criteria | Score of sub-criterion /10 | Weighting factor of sub-criterion | Score of criterion /20 | Total criteria scores /100 |
| CRITERION 1: DOCUMENTATION | 1.1 Availability of the technical documentation and other documentation related to user and maintenance instructions | 10.0 | 2 | 20.0 | |
| CRITERION 2: DISASSEMBLY, ACCESSIBILITY, TOOLS, FASTENERS | 2.1 Ease of disassembly parts from List 2 (most possible malfunctioning parts) | 10.0 | 1 | 20.0 | |
| | 2.2 Necessary tools (List 2) | 10.0 | 0.5 | | |
| | 2.3 Fasteners characteristics parts from List 1 (for function of product needed parts) and List 2 | 10.0 | 0.5 | | |
| CRITERION 3: AVAILABILITY OF SPARE PARTS | 3.1 Availability over time parts from List 2 | 10.0 | 1 | 17.5 | 97.5 |
| | 3.2 Availability over time parts from List 1 | 10.0 | 0.5 | | |
| | 3.3 Delivery time parts from List 2 | 5.0 | 0.3 | | |
| | 3.4 Delivery time parts from List 1 | 5.0 | 0.2 | | |
| CRITERION 4: PRICE OF SPARE PARTS | 4. Ratio between price of parts from list 2 to the price of the product | 10.0 | 2 | 20.0 | |
| CRITERION 5: SPECIFIC CRITERION | 5.1 Free remote assistance | 10.0 | 1 | 20.0 | |
| | 5.2 Possibility to use multi-products battery | 10.0 | 1 | | |
| Reparability index of 10 | | | | 9.8 | |

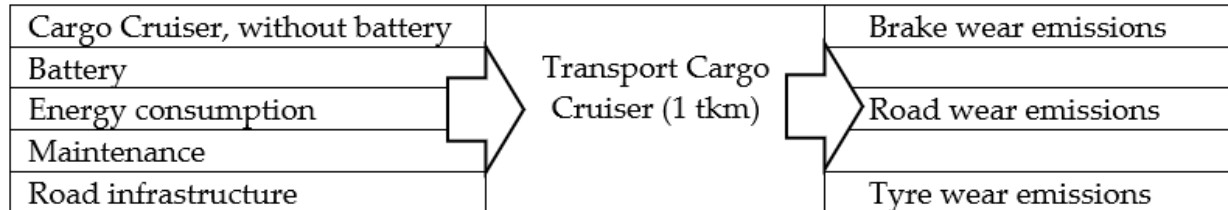

**Figure 4.** Balancing framework for Cargo Cruiser light vehicles, related to functional unit of one tonne-kilometre transport with vehicle in respective design mode.

In the target definition of LCA, different environmental impact categories were selected in order to map their possible shifts. The global warming potential according to IPCC 2013 [21] is significantly reduced for the Cargo Cruiser, using both constructions, with a 250 kg payload and 0.22 kg $CO_2$-eq/tkm compared to EV or LDV, all with payload

of 250 kg (Figure 5). The ascertained GWP meets expectations and is plausible compared to balances found in the literature [22].

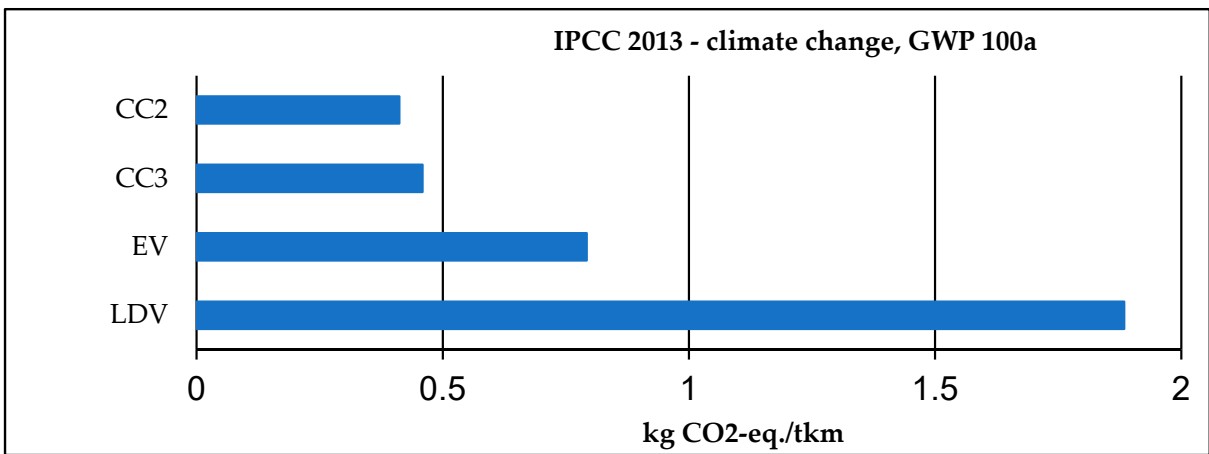

**Figure 5.** Comparison of global warming potential (GWP 100a) according to IPCC 2013, of CC2, CC3, EV and LDV. Each vehicle was loaded with 250 kg payload and balance was related to one tonne-kilometre (1 tkm).

In order to balance the GWP further, LCIA investigations was carried out to identify possible side effects. ReCiPe (Figure 6) was used because of its relevant aggregation of endpoints, which enables a comprehensive comparison of systems as vehicles in use for the last mile process.

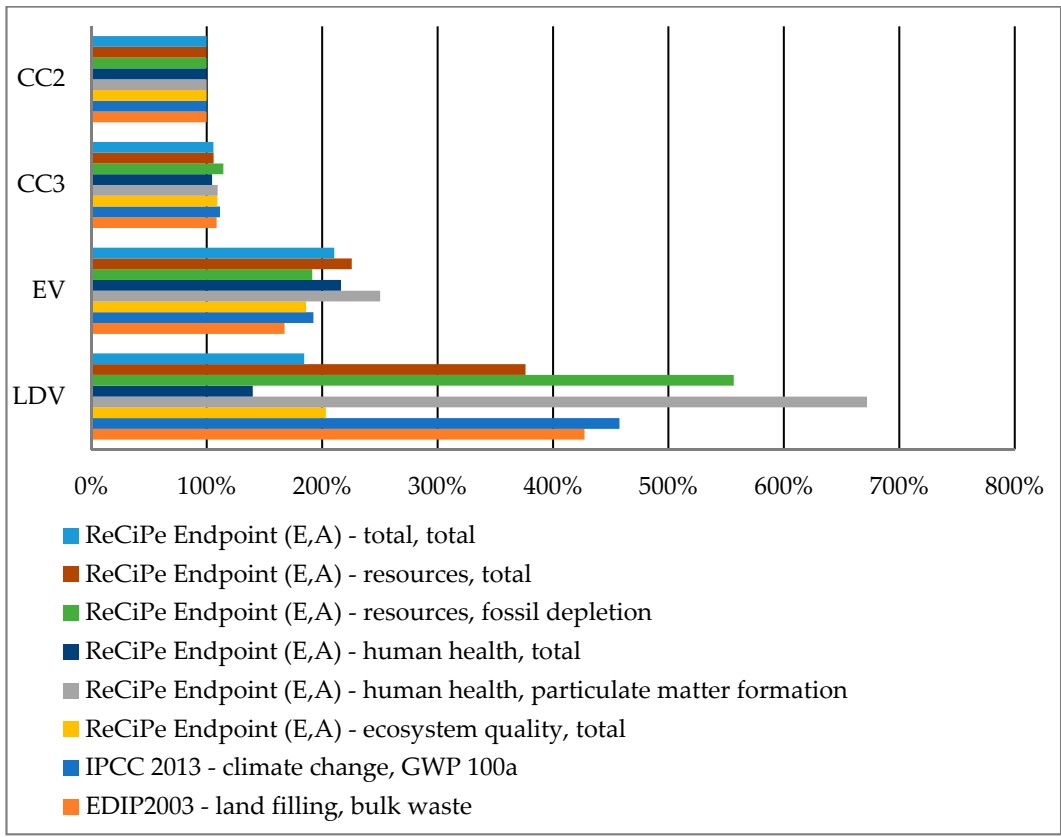

**Figure 6.** Relative comparison of transport vehicles (CC2 normalised to 100%) of IPCC 2013 GWP, ReCiPe key figures and EDIP key figure for landfilling of residual materials for CC3, EV and LDV, each with 250 kg payload, based on one tonne-kilometre (1 tkm).

Because of the larger amount of glass fibre in GFRP construction, to meet conventional estimations in mechanical design with fibre composites in terrestrial vehicles, the weight of CC3 was 3% higher than CC2. In this way, on one side, the energy need for driving (ReCiPe endpoint (E,A) resources, fossil depletion) increases, as does the environmental impact from production and end of life of GFRP (all other ReCiPe endpoints).

The EDIP 2003 indicator for land filling of bulk waste also reflects the larger amount of glass fibre composite in the GFRP construction of CC3, increasing proportional to the mass stream of GFRP.

The ReCiPe (E,A) indicator for human health and particulate matter formation increases significantly with the weight of the vehicle, which is mainly related to emissions from tires and brakes when focusing on CC and EV. When the weight of CC and EV is nearly doubled, the formation of particulate matter will more than double. When the weight of LDV is quadrupled to CC, the formation of particulate matter of LDV is more than six times higher than that of CC, which shows the relation of this ReCiPe endpoint indicator to vehicle mass and the impact of combusting diesel fuel is stronger than the use of the German electricity mix for road transport of a 250 kg payload.

Relative to CC and LDV, EV shows a strong to significant increase in ReCiPe (E,A), total human health related to production of the traction battery.

## 4. Discussion

In the cradle-to-grave LCA model, landfilling was selected for the end of life (grave) of the fibre composite components, as no primary or secondary data were available on the recycling of fibre composites and landfilling or storage at the end of their life was, therefore, assumed to be the practice. In this scenario, the use of fibre composites in vehicle construction has significant environmental impacts at the end of their life according to the EDIP 2003 indicator in comparison with conventional EVs (Figure 6). Other balanced environmental impacts reported, GWP and ReCiPe endpoints, show a significant reduction in direct relation to vehicle mass.

The weight of the LEV, which is around 50% lower than that of a conventional EV, leads to a significant reduction of around 50% in environmental impact (Figure 6), despite the conservative assumed functional unit tonne-kilometre.

The resource and energy-efficient transmission of data by LoRaWAN was validated as a robust option for transmission of vehicle data for structural health monitoring (SHM). Therefore, existing LoRaWAN networks or singular networks with related databases for last mile traffic with LEVs can be used in urban areas. In addition, structural health monitoring, this assessment of vehicle data can be used for tracking overload events that occur unexpectedly in practical operation to be registered in digital "component biographies" of vehicle elements to support approaches for reuse or extended component service life.

The French repair index is currently not directly usable for the mapping of light vehicles. The parameter set for battery-powered electric lawnmowers just allows an initial appraisal for possible repair of LEVs.

In further investigations, the assessed approaches can be developed further to improve the usability and the mass-to-weight ratio of LEVs for last mile services in personal transport and the transport of goods.

In order to make the repair index directly applicable to LEVs, extending the parameter sets to include LEV-specific components, such as mechanical steering and mechanical coupling with muscle power, is needed.

To solve the problem of fibre composite end of life, it is proposed to investigate the use of natural fibre composites with non-genotoxic resin matrices. It should be investigated how such composites can be processed at the end of their life as biomass input for pyrolysis processes such as thermocatalytic reforming (TCR) into so-called plant charcoal as a soil substrate, to reduce the negative environmental impact.

The practical use of SHM using shape memory alloy (SMA) sensors has to be validated for practical use in vehicle operation, based on the findings of LoRaWAN data transmission. The use of SMA sensors for measuring the strain on vehicle parts enables the optimisation of vehicle construction by comparing theoretical predicted loads (via FEM) with real-life measured mechanical loads on composite parts and, if necessary, detecting misuse or mechanical stresses that can occur in practical use. Based on such a proposed SHM, future investigations will adjust the safety factors for the fibre composite design to optimise the mass balance (cumulative resource expenditure) for fibre composite structures in light-vehicle construction.

In order to specifically reflect the new mass–power–weight ratio of light vehicles on the last mile, a new kind of functional unit, "last mile delivery of standard parcels", is proposed for future LCA of parcel delivery on the last mile with LEVs and electrically supported bicycles.

**Author Contributions:** The conceptualization of the investigation was by S.W., O.L. and A.F. equally. The LCA modelling was carried out by S.W. and P.B. Writing and original draft preparation were by S.W. All authors have read and agreed to the published version of the manuscript.

**Funding:** This research was funded by German Federal Ministry of Education and Research (BMBF), grant number 033R245A (LEVmodular project, ReziProK programme).

**Data Availability Statement:** Primary data used in this research came from internal operational data of the project partners. Secondary data used in this study came from the proprietary Ecoinvent v3 LCI database.

**Acknowledgments:** The authors thank the Federal Ministry of Education and Research (BMBF) for funding the research work within the funding programme "Ressourceneffiziente Kreislaufwirtschaft—Innovative Produktkreisläufe (ReziProK)" under grant number 033R245A and the three anonymous reviewers for their constructive advice.

**Conflicts of Interest:** The authors declare no conflict of interest. The funders had no role in the design of the study; in the collection, analyses, or interpretation of data; in the writing of the manuscript; or in the decision to publish the results.

## Abbreviations

| | |
|---|---|
| CC2 | Cargo Cruiser 2 (light electric vehicle with steel/GFRP construction) |
| CC3 | Cargo Cruiser 3 (light electric vehicle with GFRP construction) |
| EDIP | Environmental Development of Industrial Products |
| EV | Electric vehicle |
| FEM | Finite element method |
| GF | Glass fibre |
| GFRP | Glass fibre reinforced plastic |
| GWP | Global warming potential |
| LCA | Life cycle assessment |
| LDV | Light-duty vehicle |
| LEV | Light electric vehicle |
| NF | Natural fibre |
| NFRP | Natural fibre reinforced plastic |
| SHM | Structural health monitoring |
| SMA | Shape memory alloy |
| SME | Small and medium-sized enterprises |

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
