# Peer review of "Light Electric Vehicles for Muscle–Battery Electric Mobility in Circular Economy: A Comprehensive Study"

_sustainability, doi:10.3390/su132413793_

Round 1

Reviewer 1 Report

The general idea of the paper seems to be good. However, the paper organization can  be improve and there are several minor technical challenges that should be effectively addressed.

  1. My suggestion is to divide the introduction into three subsections: 1) motivation and incitement, 2) literature review and 3) contribution and paper organization
  2. Please compare your Light Electric Vehicle Cargo Cruiser with other solutions with more detail (table 4)
  3. It is recommended to provide the list of abbreviations at the end of the paper.

Author Response

Thank you for suggestions!
1) will divide the introduction
2) table 4 will be added with LDV (then 3 type of vehicle will be compared)
3) List of abbrevations will be included

Reviewer 2 Report

In this paper, two construction methods are used to design the whole vehicle structure, and comparative inspection and evaluation are carried out in the whole life cycle. The resource efficiency of the light electric vehicle is proved from the ecological point of view. On the whole, it is a meaningful and thoughtful paper. After I go through the whole manuscript, this paper would be suitable for publication in the SUSTAINABILITY journal after some significant modifications due to the following questions/comments.

Major concerns:

  1. Do not cite references in the abstract.
  2. The Introduction should be re-organized so that the test flows in a logical manner from problem description to problem constraints and interactions of these constraints, to proposed methodology for optimally solving the problem, computational and space complexity as well as solution accuracy, comparisons with existing - as well as alternative methodologies (again in terms of computational and space complexity as well as solution accuracy), potential extensions and future directions.
  3. The paper does not specifically describe the light vehicle model used. Please indicate the parameters of the vehicle and the model.
  4. The methods and materials used in the paper are very sketchy. Please fill in the details of this section.
  5. The resolution of some figures needs to be improved to meet the publishing requirements, such as Table. 1, ^. You may have used different formats when generating the article.
  6. Table 2 appears to be a screenshot of the software interface. Please redraw the picture for better aesthetics and resolution.
  7. A large number of images in the text have grey edges.
  8. Nice experiments are done. However, the practical implications of the results should be discussed in more detail. Moreover, you should explain the applicability of the proposed methodology and provide useful managerial insights.
  9. Chapter 3 is rambling and illogical. What is the connection between them? Authors should strengthen the linkages between studies.
  10. The studies in Tables 3 and 4 are more like research results. What practical significance do they have on the test vehicle?
  11. Define Nomenclature, Greek symbols, subscripts, superscripts, and acronyms separately in table form.

The article is too short. The research content is much, but the description is not rigorous enough.

Author Response

Thank you for the constructive advising. Please see the here the changelog:

  • Introduction is extended (description of vehicle)
  • Figures are redesigned
  • Introduction into 3 is given (to bring themes together)
  • Table 4 is extended and interpreted
  • Conclusion is extended and furthe works proposed
  • Abbrevation is inserted
  • Literature list is updated

Reviewer 3 Report

Authors present a vehicle configuration was developed which, in addition to resource-saving production and longlife operation. The paper is very well written, according to all academic standards. The overall rating of the paper is good.

I have the following comments and remarks:

  1. What the paper might lack is a clearer statement of how the model differs from previous works and thus contributes to the literature. Which are the main contributions and the novel characteristics of the proposed methodology to the scientific community? Is there a novel methodological framework, or at least some enhancements to an existing one? The authors should clarify those aspects.
  2. In section 1, please include a comparative table that give a better understanding of different approaches that address the relationship between approaches of electric cars

  1. The abstract needs to be improved. The authors are encouraged to highlight the key findings of the work as well as some comments for the applicability of the adopted approach in the field of electric vehicles.

  1. The paper lacks a description of the model validation procedure (e.g., model intercomparison). I believe it would be beneficial for the paper to include it in its final version.

  1. Could you please elaborate more (mainly in the conclusions section) on the model limitations from a methodological viewpoint that need to be addressed by future works?

Author Response

(The authors gave the same response as above.)

Round 2

Reviewer 2 Report

In this paper, two construction methods are used to design the whole vehicle structure, and comparative inspection and evaluation are carried out in the whole life cycle. The resource efficiency of light electric vehicle is proved from the ecological point of view. After a revision, the article became rich in content. However, the author's reply is not detailed enough. After I go through the whole manuscript, this paper would be suitable after some minor modifications due to the following questions/comments.

  • The author's reply to the review opinion is not rigorous enough, please reorganize the language.
  • Do not cite references in the abstract.
  • The methods and materials used in the second chapter are very rough. Please fill in the details in this section.
  • What is the practical significance of the analysis in FIG. 2 (a)(b)? This part is not specifically introduced in the paper, so the author is requested to analyze the significance of this part of the research and the correctness of the finite element analysis in detail.
  • The picture quality in Table 4 is not clear

Author Response

Hello,

here short change-report:

- language service will be choosen

-references in abstract deleted

- details are filled in 2nd chapter, but hope it will not go into detail so much, of material-development etc., because aim of this article is a comprehensive view

- old Fig 2 (a)(b) is changed and moved to 2nd chapter. FEM now is validated by real world test (pull out test of safety belts successfully as expected)

- Table 4 is cleared

--

Reviewer 3 Report

The authors have satisfactorily responded to all my questions and made the necessary changes to the manuscript. 

Author Response

thank you for your input and as info: some further changes done in relation to review person 2 

--